# Dual-Modular Versus Single-Modular Stems for Primary Total Hip Arthroplasty: A Long-Term Survival Analysis

**DOI:** 10.3390/medicina59020290

**Published:** 2023-02-02

**Authors:** Samo K. Fokter, Nejc Noč, Vesna Levašič, Marko Hanc, Jan Zajc

**Affiliations:** 1Department of Orthopaedics, University Medical Centre, 2000 Maribor, Slovenia; 2Faculty of Medicine, University of Maribor, 2000 Maribor, Slovenia; 3The National Arthroplasty Registry of Slovenia, Valdoltra Orthopaedic Hospital, 6280 Ankaran, Slovenia; 4Department of Orthopaedics, General Hospital, 9000 Murska Sobota, Slovenia

**Keywords:** total hip arthroplasty, modular necks, exchangeable necks, survival rate, complications, revision rate, primary hip replacement, prosthesis, Ti-alloy

## Abstract

*Background and Objectives*: Increased revision rate of dual-modular (DM) femoral stems in primary total hip arthroplasty (THA) because of modular-neck breakage and adverse local tissue reactions (ALTRs) to additional junction damage products is well established and some designs have been recalled from the market. However, some long-term studies of specific DM stems did not confirm the inferiority of these stems compared to standard single-modular (SM) stems, and a head-to-head comparison THA is missing. The objectives of this multicentre study were to determine the survivorship and complication rates of a common DM stem design compared to a similar SM stem. *Materials and Methods*: In a time frame from January 2012 to November 2015, a cohort of 807 patients (882 hips) consecutively underwent primary cementless THAs at two orthopaedic centres. 377 hips were treated with a Zweimüller-type DM stem THA system and 505 hips with a similar SM stem THA system, both including a modern press-fit acetabulum. Kaplan-Meier survivorship and complication rates were compared between both groups in a median follow-up of 9.0 years (maximum, 9.9 years). *Results*: The 9-year survivorship of the DM stem THA system (92.6%, 95% CI 89.9–95.3) was significantly lower than that of the SM stem THA system (97.0%, 95% CI 95.2–98.8). There were no differences in revision rates for septic loosening, dislocation, and periprosthetic fractures between the two groups. One ceramic inlay and one Ti-alloy modular neck breakage occurred in the DM stem THA system group, but the main reason for revision in this group was aseptic loosening of components. *Conclusions*: The survivorship of the DM stem THA system was lower than the similar SM stem THA system in a comparable clinical environment with long-term follow-up. Our results confirmed that no rationale for stem modularity exists in primary THAs.

## 1. Introduction

Orthopedic surgeons have traditionally been using “monoblock” femoral stems in total hip arthroplasty (THA) to treat progressive pain and disfunction of patients with ostoarthritis and other hip joint diseases. Soon, most commercially available THA systems were modernized with modular heads with different diameters, bore lengths and material properties, i.e., the single-modular (SM) femoral stems. The dual-modular (DM) femoral stem is an innovation in primary THA claimed to offer the surgeon more choice in controlling femoral offset, leg length, and hip stability [1]. Better re-establishing of appropriate hip biomechanics and added neck-stem modularity were believed to slow implant wear and–via temporary removal of the interchangeable neck–ease the accetabular component revision, respectively. While some earlier studies have proven the DM concept showing better restauration of anatomical conditions, other authors found no clear differences regarding hip geometry and stability compared to standard SM femoral stems [2,3]. Besides, an increased number of studies have described catastrophic failures of the modular necks because of fretting corrosion at the stem-neck junction, i.e., modular neck breakage [4,5,6,7,8]. Crevice corrosion usually made disconnecting the neck-stem junction impossible during revision. Another concern is the possible local adverse tissue reactions (ALTRs) from the additional modular junction metal debris [9]. These damage products of the neck-stem junction might be responsible for the increased revision rate of DM hip systems shown by some national arthroplasty registries [10,11].

Several prospective and retrospective studies evaluated the risk of revision of the primary THA using DM stem prostheses [12]. However, only two single-centre studies included a comparator group of patients who underwent SM stem THA [3,13]. To the best of our knowledge, no long-term study comparing the survivorship and behaviour of a DM femoral stem THA system with otherwise similar SM femoral stem THA system was published to date. The aim of this exploratory retrospective multicentre study is to report on the overall long-term survival rate of DM stem THA system compared to a similar SM stem THA system in a series of patients treated for common indications in anatomically normal hip joints.

## 2. Materials and Methods

### 2.1. Patient Characteristics

This study integrates orthopaedic patients who have undergone primary uncemented THA for usual indications in two orthopaedic centres (one community hospital and one university setting). Both centres are part of the national public health care system. All consecutive patients treated with one of the studied THA systems between January 2012 and November 2015 were included. This time period was chosen because the studied THA systems were available at the surgeons’ preference in both centres. The data about primary uncemented THAs was recovered from the central archives of each institution, which were only partially organized in the digital mode. From the archives, the demographic data and the implantation data were extracted. In this way, 807 patients with 882 hips were available for further analysis: 377 hips were treated with a DM stem and 505 hips were treated with an SM stem. There were 75 patients (9.3%) in the cohort who have had THA performed on both hips. Out of the 75 patients, there were 38 (50.7%) patients with the DM stem implanted on both sides, 35 (46.7%) patients with the SM stem implanted on both sides, and 2 (2.6%) patients with both an SM stem implanted on the left side and a DM stem implanted on the right side.

Patients treated with a THA for an acute proximal femoral fracture, patients with significant anatomical deformities of the hip, patients with rheumatoid arthritis, patients with renal disease, and patients treated with other commercially available THA systems were excluded.

Through the help of The National Arthroplasty Registry of Slovenia (NARS), data about the revisions of specific patients and data about all the patients who have passed away during the window of observation, which lasts from 1st of January 2012 until 15th of November 2021, were evaluated. Revision after the primary THA represents the endpoint of this study and defines the survival of the specific hip implant system.

### 2.2. Surgery Characteristics

All surgeries were performed by 5 skilled orthopaedic surgeons with more than 5 years of experience in THA, performing at least 100 THAs yearly. A standard anterolateral approach was used in all cases. Preoperative planning was performed using analogue templates on standard calibrated 110-cm AP pelvic X-rays. Intraoperative cup positioning was established with an aiming device and checked with the Ranawat sign. This way, hip anatomy was restored, hip instability prevented, and leg length discrepancy avoided to the best possible extent.

Patients received routine follow-up at institutional outpatient clinics at 3 months, 6 months, and 1 year postoperatively and thereafter on request of the patient and/or his or her family physician.

### 2.3. Implant Characteristics

The Profemur Z stem (Wright Medical Technology, Arlington, TN, USA) is a DM dual-tapered grid-blasted titanium alloy (Ti6Al4V) femoral stem that closely follows the original Zweimüller’s rectangular single-modular design. The DM stem was sold as a part of an uncemented THA system, which included a titanium-alloy press-fit acetabular cup with ultra-heavy-weight-polyethylene (UHMWPE) or alumina matrix Zirconia composite (Biolox Delta, CeramTec GmbH, Plochlingen, Germany) inlay (Procotyl L, Wright Medical Technology). The Alloclasic SL stem (Zimmer, Warshaw, IN, USA) is an SM dual-tapered grid-blasted titanium alloy (Ti6Al4V) femoral stem of the original Zweimüller design. The SM stem was sold as a part of an uncemented THA system, which included a titanium-alloy press-fit acetabular cup with UHMWPE or alumina matrix Zirconia composite (Biolox Delta, CeramTec) inlay (Allofit, Zimmer). Since the studied DM stem did not have an SM option and vice versa, the chosen SM stem had similar geometry to reduce bias from different stem designs.

### 2.4. Statistical Analysis

Data were processed with SPSS software (version 27.0, SPSS Inc., Chicago, IL, USA) on a personal computer. The survival analysis was performed, with revision of any component for any reason as the endpoint, using the Kaplan–Meier test (with 95% confidence interval). To compare the Kaplan-Meier curves, the Log Rank (Mantel-Cox) test was used. Statistical significance was a *p* value of less than 0.001. The survival times of unrevised implants were censored at the last observation (patient death or 15 November 2021, whichever came first). The categorical data were expressed as a number with percentages and continuous data were expressed as median with 95 percent confidence interval. Nominal data were compared with Chi-Square test and numerical continuous data with Mann-Whitney U-test, with a level of significance of 0.05 used in both tests.

## 3. Results

518 surgeries were performed on a tertiary academic hospital and 364 on a community hospital level. Demographic data are shown in Table 1.

The BMI subgroups (underweight <19 kg/m^2^, normal 19–24.9 kg/m^2^, overweight 25–29.9 kg/m^2^, obese 30–34.9 kg/m^2^, and morbidly obese >= 35 kg/m^2^) were equally distributed between both groups (*p* > 0.05). The data is shown in detail in Figure 1.

Most of the patients in the SM stem THA system group and in the DM stem THA system group were treated for primary OA. Significantly more patients with idiopathic femoral head necrosis and patients with developmental dysplasia of the hip were treated in the SM stem THA system group. The indication for THA is shown in Table 2.

Only femoral heads with diameters 28 mm, 32 mm and 36 mm were used. No metal-on-metal articulation was used. The implant characteristics are shown in detail in Table 3.

Of the 377 modular necks inserted in combination with DM femoral stem, 215 (57.0%) were made of Ti6Al4V alloy, 161 (42.7%) of Cobalt-Chromium (Co-Cr) alloy, and no data about neck material existed in 1 (0.3%) case. The data of femoral stem size was not provided for 8 (0.90%) hips, sizes were equally distributed between both groups (*p* > 0.05), and the data is shown in detail in Figure 2.

The median follow-up for SM stem system was 8.0 years (95% CI 8.0–8.0) and 9.0 years (95% CI 9.0–9.0) for DM stem system (*p* < 0.001). During the observational period, 174 patients have died. The maximum follow-up was 9.9 years for all patients who were still alive on 15 November 2021. The 9-year survival rate for any reason including aseptic loosening of any component of the DM stem THA system (92.6%, 95% CI 89.9–95.3) was significantly lower than that of the SM stem THA system (97.0%, 95% CI 95.2–98.8). The Kaplan-Meier survival curves with 95% confidence interval for DM and SM THA system, respectively, are shown in Figure 3.

There were 13 cases of revision in the SM stem THA system group and 30 cases of revision in the DM stem THA system group. The median time to revision was 1.0 years (95% CI 0.0–4.0) for SM stem THA system and 0.5 years (95% CI 0.0–2.5) for DM stem THA system (*p* > 0.05). Aseptic loosening was the only statistically significant reason for revision between the groups in favor of DM stem THA system (*p* < 0.05). The 1 modular neck breakage occurred in a modular neck made of Ti-alloy in a 60-year-old male with a BMI of 30.4 kg/m^2^ 6.5 years after primary implantation. In the DM stem THA system, 1 dislocation occurred with a 36 mm head diameter, in the SM stem THA system 1 dislocation occurred with a 28 mm head diameter, and 1 additional dislocation occurred in each group with a 32 mm inserted head diameter. The other reasons for revisions are reported in detail in Table 4.

## 4. Discussion

Our results show that better long-term results could be expected after THA with SM femoral stem system compared to THA with DM femoral stem system.

A recent study using the data from an Italian hip replacement registry analysed 2857 DM femoral stem THA performed between January 2000 and December 2009 [14]. The patients were treated with AncaFit (Cremascoli Ortho, S.r.l, Milan, Italy) hip prosthesis, the forerunner of DM stem THA system (Wright Medical Technology) investigated in this study with identical neck-stem coupling. Only complete AncaFit THA systems with the subject stem and cup identified by product code, Ti-alloy/Ti-alloy neck-stem junction, and all available articulation couples were included in the analysis. The authors found that the 17-year survival rates for any reason and aseptic loosening of any components were 88.9%. With only 2 cases of modular neck breakage and one case of ALTR identified, the authors concluded that the risk of modular neck breakage is lower than previously reported and that the incidence of neck-stem junction damage products does not significantly alter the rate of implant loosening [14].

Similar to Baleani et al., we have also focused on the complete hip prosthesis, i.e., all implant, as has been sold by the manufacturer as the “hip system” with the subject stem, head, cup and liner identified by product codes. In this way, hybrid implants were excluded due to potential bias in clinical outcomes from association of components of different prostheses. It has been shown by Chana et al. that corrosion products from taper caused by mismatch of the implant components lead to pseudotumor formation requiring revision [15]. Pseudotumor formation can also occur within original DM femoral stem THA systems and could be extensive when pairing dissimilar metals. A reaction to metal debris coming from a neck-stem junction can remain misdiagnosed for a long time, as it was described by Canella et al. [16]. These authors reported on a case of a 73-year-old woman who ambulated with a progressive right hip pain for over a year before the correct diagnosis was established. During the two-stage revision the authors confirmed a 20 × 10 cm pseudotumor formation [16].

In our study the incidence of modular neck fracture was also lower than previously reported in a national survey [17]. On the other hand, higher revision rate of DM stem THA system compared to a similar SM stem THA system was confirmed because of aseptic loosening. This contrasts with the study of Baleani et al., which did not include data of a suitable comparator. However, only 49.8% of the patients in the study of Baleani et al. were overweight or obese, while 77.7% of the patients receiving DM stem THA system and 78.1% of the entire cohort in the present study was overweight, obese, or morbidly obese. It has been shown previously that increased BMI is associated with increased incidence of modular neck fracture [18,19,20].

In a retrospective cohort study, Laubscher et al. recently compared the results of arthroplasty procedures performed at a district and at a tertiary academic hospital, respectively [21]. Our results are in agreement with these authors who concluded that arthroplasty at district health care level is safe and may reduce the pressure on arthroplasty services at tertiary care facilities. In the present study, we have also included data from a community hospital and a tertiary academic centre to avoid limiting the generalization of the results. Namely, in the previous studies only data from major orthopaedic centres were included and no comparable SM stem THA system was analysed [17,22]. While in both mentioned studies the incidence of modular neck fractures was about 1%, in the present study the incidence was less than 0.3% (1/353 hips with DM stems). This may be due to the relatively small number of Co-Cr necks (42.7%) used in the present series. It has been previously reported that Co-Cr modular necks in combination with Ti6Al4Ti femoral stems fracture earlier than Ti6Al4Ti modular necks and can lead to release of metal ions and debris resulting in local soft-tissue destruction [23,24,25,26,27]. Even though no formal data of ALTRs were found in the patients’ records, the higher revision rates of DM stem THA system because of aseptic loosening may be the consequence of additional modularity at the stem-neck junction.

To the best of our knowledge, this is the first long term head-to-head study comparing modular and comparable nonmodular femoral implants in primary THA. Duwelius et al. have studied 284 patients receiving a primary uncemented THA with a nonmodular femoral stem (M/L Taper, Zimmer, Warsaw, IN, USA) and 594 patients receiving a modular femoral stem (M/L Taper Kinectiv, Zimmer) [13]. In the single-surgeon study with a 2.4-year mean follow-up, the authors concluded that the use of modular neck stems did not reduce the likelihood of revision nor improve hip scores. Regarding the revision as the endpoint, our results have confirmed inferior long-term results of the DM stem THA system compared to a similar uncemented modern SM stem THA system.

Gerhardt et al. have studied restoration of hip geometry on preoperative and postoperative calibrated radiographs in 95 consecutive primary THAs with a modular neck design (Profemur Z, Wright Medical) and compared them with 95 match-controlled THAs with a similar monoblock stem (Alloclassic, Zimmer) [3]. Within one-year follow-up, the authors concluded that modular necks did not reveal a clear benefit in restoring hip geometry and dislocation rate compared to straightforward THA. In our series, we have also compared consecutive primary THAs with a common DM femoral stem design (Profemur Z, Wright Medical) with a similar SM femoral stem (Alloclassic, Zimmer) because the overall geometry of the stems was grossly comparable. Both stems were namely of a rectangular cross-section, dual taper geometry with a heavy grit blast surface finish and had a similar 12/14 Morse taper. In accordance with Turley et al., we agree with the aforementioned authors that in the light of potential disadvantages, including catastrophic modular neck failures, the presumed advantages of the DM stems should not be overestimated [28].

Several studies have shown satisfactory results of DM stem THAs [29,30]. However, these were only short to middle term single-centre observational studies with no control group. Besides, a review of a local registry data has tried to prove noninferiority of a cementless modular stem (Profemur Z, MicroPort Orthopedics Inc., Arlington, TN, USA) compared to a bunch of contemporary fixed neck uncemented stems in the long term [31]. The authors have shown that the 12-year survivorship of the DM stem (95.8%) was not significantly lower than that of all fixed neck stems (96.1%). It is not insignificant to mention that the first author of that study was an employee of the MicroPort Orthopedic Inc., the successor of Wright Medical Technology, Inc. However, the company did not apply for the CE mark in 2022 and their DM stems for THA are no longer available on the European market. Larger national registries data have clearly shown that DM stems perform significantly worse than uncemented monoblock stems in THA [10,11].

We acknowledge certain limitations in this study. Firstly, the patients in the DM stem THA group were younger than the patients in the SM stem THA group. Age of the patients determines their activity level, which could influence the survival of the THA. Thus, our results are potentially biased. Secondly, no data existed about the functional outcomes after THA in our retrospective series in which the revision surgery was considered as a prosthesis failure. Thirdly, since the shape of proximal Morse taper of modular neck is symmetrically elliptical and postoperative X-ray analysis were not performed in this study, we have no data on the actual position of 33.5% of the implanted DM stems. E.g., a Varus-Valgus neck could be inserted in varus or in valgus position intraoperatively. This position could theoretically affect performance of the entire THA system. Fourthly, aseptic loosening was the major reason of revision in the DM stem group. Some of these revisions may actually represent the consequences of the ALTR. Since no routine magnetic resonance imaging (MRI) was performed before revision in these patients, we were not able to prove that. Fifthly, not all patients’ data were available for analysis because of poor data acquisition and storage in both medical institutions.

## 5. Conclusions

Our results confirmed that no rationale for stem modularity exists in primary THAs. The survivorship of the DM stem THA was lower than the similar SM stem THA in a comparable clinical environment with long-term follow-up. The incidence of modular neck breakage was lower than previously reported. Early aseptic loosening–and not modular neck fracture–represented the major single reason for revision in the DM stem THA system group. However, since the patients in the DM stem THA system group were younger than patients in the SM stem THA system group, our results should be taken with caution. Besides, patients who required revision in our series were not checked with metal artefact reduction sequence (MARS) MRI to detect possible ALTR. Close monitoring is required in patients treated with DM stems reported in the present study and other primary DM stem THA systems still available in the orthopaedic armamentarium.

## Figures and Tables

**Figure 1 medicina-59-00290-f001:**
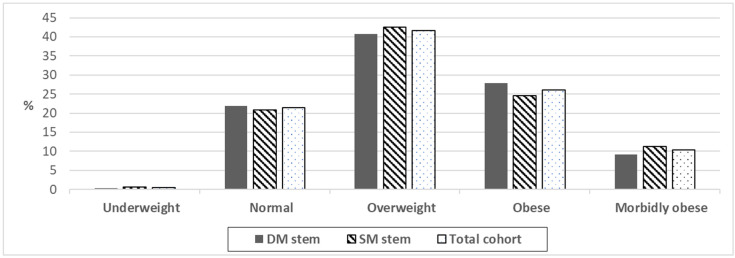
BMI subgroups distribution between the groups and of the total cohort. DM–dual-modular stem, SM–single-modular stem.

**Figure 2 medicina-59-00290-f002:**
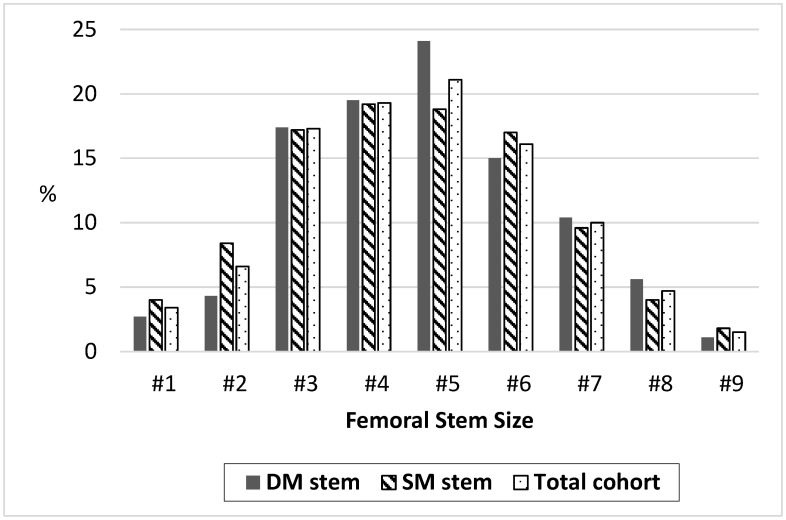
Femoral stem size distribution between the groups and of the total cohort (proportion from valid). DM–dual-modular stem, SM–single-modular stem.

**Figure 3 medicina-59-00290-f003:**
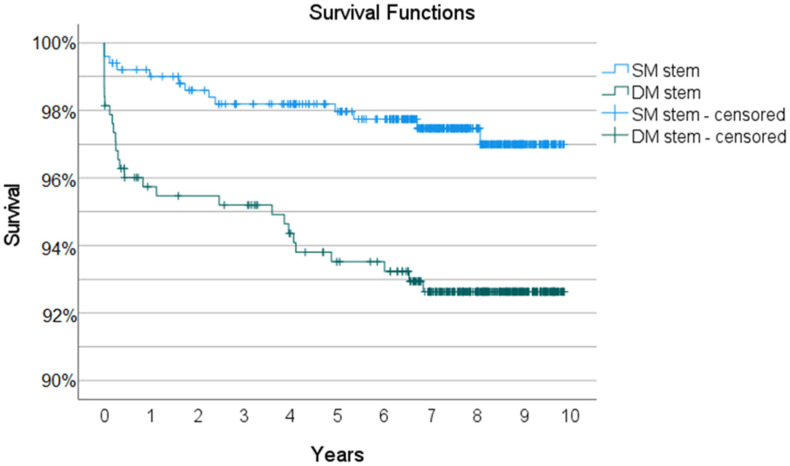
Kaplan-Meier survival curves with 95% confidence interval. DM–dual-modular stem, SM–single-modular stem.

**Table 1 medicina-59-00290-t001:** Demographic data of the cohort.

	DM Stem	SM Stem	Total Cohort	*p*-Value
Number of THA implanted (%)	377 (42.7)	505 (57.3)	882 (100)	
Gender (male/female; N (%))	163/214 (43.2/56.8)	196/309 (38.8/61.2)	359/523 (40.7/59.3)	0.186
Age at implantation (years; median and 95% CI for median)	68.0 (67.0–69.0)	71.0 (69.0–72.0)	69.0 (69.0–71.0)	<0.05
Side * (left/right; N (%^1^))	163/196 (45.4/54.6)	182/231 (44.1/55.9)	345/427 (44.7/55.3)	0.710
Weight ** (kg; mean and 95% CI)	79.56 (78.17–80.94)	79.69 (78.09–81.38)	79.63 (78.53–80.72)	0.558
BMI *** (kg/m^2^; median and 95% CI for median)	28.41 (27.68–29.07)	28.01 (27.64–28.73)	28.37 (27.76–28.73)	0.821

N–count; %^1^–proportion
from valid; * The data of the side was not provided for 110 (12.5%) hips; ** The data for patient’s weight was not provided for 123 (13.9%) hips; *** The data for BMI was not provided for 124 (14.1%) hips. DM–dual-modular stem, SM–single-modular stem.

**Table 2 medicina-59-00290-t002:** Indication for total hip arthroplasty.

	DM Stem N (%^1^)	SM Stem N (%^1^)	Total Cohort N (%^1^)	*p*-Value
Primary osteoarthritis	319 (90.4)	330 (79.7)	649 (84.6)	<0.05
Idiopathic femoral head necrosis	14 (4.0)	31 (7.5)	45 (5.9)	<0.05
Developmental dysplasia	3 (0.8)	40 (9.7)	43 (5.6)	<0.05
Femoral neck fracture sequelae	9 (2.5)	8 (1.9)	17 (2.2)	0.563
Other	8 (2.3)	5 (1.2)	13 (1.7)	0.258

N–count; %^1^–proportion from valid; data of indication for primary THA was not provided for 115 (13.04%) hips. DM–dual-modular stem, SM–single-modular stem.

**Table 3 medicina-59-00290-t003:** Implant characteristics of the femoral stems, heads and modular necks used.

	DM Stem (%)	SM Stem (%)	Total Cohort (%)
Head diameter			
28 mm	8.8	25.4	18.3
32 mm	42.8	12.9	25.6
36 mm	48.4	61.7	56.0
Head length			
S (−3.5 mm)	33.7	10.3	20.2
M (0 mm)	34.5	32.5	33.4
L (+3.5 mm)	30.2	42.3	37.1
XL (+7.0 mm)	1.6	14.9	9.1
Articulation type			
CoP	28.9	55.2	44.1
MoP	8.5	23.4	17.2
CoC	62.6	21.4	38.7
Modular neck orientation and length			
Straight–short	50.5	NA	-
Straight–long	16.0	NA	-
Varus-Valgus–short	17.8	NA	-
Varus-Valgus–long	8.2	NA	-
Ante-Retro–short	2.1	NA	-
Ante-Retro–long	3.2	NA	-
Varus-Valgus/Ante-Retro–short	0.8	NA	-
Varus-Valgus/Ante-Retro–long	1.3	NA	-

CoP–ceramics-on-polyethylene; MoP–metal-on-polyethylene; CoC– ceramic-on-polyethylene; NA–not applicable; DM–dual-modular stem; SM–single-modular stem.

**Table 4 medicina-59-00290-t004:** Indication for revision surgery.

	DM Stem N (%)	SM Stem N (%)	Total Cohort N (%)	*p*-Value
Infection	8 (26.7)	3 (23.1)	11 (25.6)	0.804
Aseptic loosening	9 (30.0)	0 (0.0)	9 (20.9)	0.026
Periprosthetic fracture	7 (23.3)	5 (38.5)	12 (27.9)	0.310
Dislocation	2 (6.7)	2 (15.4)	4 (9.3)	0.366
Breakage of ceramic inlay	1 (3.3)	0 (0.0)	1 (2.3)	0.505
Breakage of modular neck	1 (3.3)	0 (0.0)	1 (2.3)	0.505
Other	2 (6.7)	3 (23.1)	5 (11.6)	0.123
Total	30 (100.0)	13 (100.0)	43 (100.0)	0.210

N–count; DM–dual-modular stem; SM–single-modular stem.

## Data Availability

Not applicable.

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
