# Peer review of "Dual-Modular Versus Single-Modular Stems for Primary Total Hip Arthroplasty: A Long-Term Survival Analysis"

_medicina, 2023, doi:10.3390/medicina59020290_

Round 1

Reviewer 1 Report

The manuscript is quite well prepared and may be considered for publication. However, the conclusions are rather clumsy. This manuscript should be revised before further processing. 

Title, Abstract”

-Conclusion in Abstract does not match very well with Conclusion Section.

“1. Introduction”

- Introduction is relatively too short for literature review.

“2. Materials and Methods”

-No comments.

“3. Results and Discussion”

-In Fig. 3 the title of y-axis should be SURVIVAL.

-In page 8 start from line 271 it was stated that there are some limitations. These limitations have to be stated in conclusion.

“4. Conclusions”

-Check again with the results and the abstract.

Reviewer 2 Report

The study deals a very current and relevant issue in daily clinical practice. The most interesting part concern the aim to compare, in a long-term analysis, the DM stem THA system to a similar TM stem THA system in terms of outcome, survival and complication rates.

The abstract and the introduction are comprehensive, and the purpose of the study is well reported. Furthermore, all the references included are relevant.

The methodological approach is correct and complete: all data are widely described and accurately analyzed.

The results are properly reported and summarized in illustrative and intuitive tables. Authors should better explain, if the available data allow it, how and how many times patients did checks during follow up.

Discussion is well presented: among the lines 202-209, if the authors consider it appropriate, they could add https://doi.org/10.3390/jfb13030145 to better talk about the DM stem related complications.

The conclusions are in line with the obtained results. Further studies could be useful to overcome the limitations of the study.

The study might be worthy of publication after a minor revision.

A native English speaker might be useful to proofread the text to improve the grammatical content.

Reviewer 3 Report

Fokter et al. present an exploratory retrospective multicentre study reporting the overall long-term survival rate of DM stem THA system compared to a similar SM stem THA system. The authors do a nice job presenting the background and stating clearly the objective of their work in this manuscript. They are also describing well the materials and methods used while presenting clearly their findings and conclusions. This reviewer is quite positive. There are, however, some points that need to be altered as written in more detail below.

Line 65 (and later on) please remove the gap between “9.3” and “%”. It should be written as following: “9.3%”. Please repeat that change for all the next cases that you report a percentage in this format.

Figure 3: overall image quality is poor and the reader had difficulty seeing the difference between the 4 different cases shown. Also, since there is no data corresponding to survival percentages lower than 90%, it would be nice to update the plot with the y axis (survival) lower limit being 90%. In that way the survival functions will be shown in more detail and the plot will be clearer to the reader. Last, please change the formal of the y-axis title “Survival”. Instead of writing the letters in a vertical format, write them horizontally and the rotate the whole word 90 degrees, following the format that you have on the x-axis.

Lines 185-186: Something is going wrong there with the N and N-count. The N overlap with the line and it’s not clear to the reader what these two mean as they appear.

The authors do a very nice job in the Discussion section, where they discuss their results as well as what previous studies have found. The limitations paragraph is also well-written. The Conclusions section may need to be a little bit stronger by elaborating a little bit more on the findings and the outcomes, so in that way the conclusion of the manuscript will be stronger and more complete.

Round 2

Reviewer 1 Report

The revised version is improving and the manuscript may be considered for publication. Quality of the results presentation and discussion may be increased.
